# Analysis of Intracranial Aneurysm Haemodynamics Altered by Wall Movement

**DOI:** 10.3390/bioengineering11030269

**Published:** 2024-03-09

**Authors:** Aurèle Goetz, Pablo Jeken-Rico, Yves Chau, Jacques Sédat, Aurélien Larcher, Elie Hachem

**Affiliations:** 1Computing and Fluids Research Group, CEMEF, Mines Paris PSL, 06904 Sophia Antipolis, France; pablo.jeken_rico@minesparis.psl.eu (P.J.-R.); aurelien.larcher@minesparis.psl.eu (A.L.); elie.hachem@minesparis.psl.eu (E.H.); 2Department of Neuro-Interventional and Vascular Interventional, University Hospital of Nice, 06000 Nice, France

**Keywords:** intracranial aneurysm, haemodynamics, fluid–structure interaction, arterial tissue modelling

## Abstract

Computational fluid dynamics is intensively used to deepen our understanding of aneurysm growth and rupture in an attempt to support physicians during therapy planning. Numerous studies assumed fully rigid vessel walls in their simulations, whose sole haemodynamics may fail to provide a satisfactory criterion for rupture risk assessment. Moreover, direct in vivo observations of intracranial aneurysm pulsation were recently reported, encouraging the development of fluid–structure interaction for their modelling and for new assessments. In this work, we describe a new fluid–structure interaction functional setting for the careful evaluation of different aneurysm shapes. The configurations consist of three real aneurysm domes positioned on a toroidal channel. All geometric features, employed meshes, flow quantities, comparisons with the rigid wall model and corresponding plots are provided for the sake of reproducibility. The results emphasise the alteration of flow patterns and haemodynamic descriptors when wall deformations were taken into account compared with a standard rigid wall approach, thereby underlining the impact of fluid–structure interaction modelling.

## 1. Introduction

Intracranial aneurysms (IAs) are pathological dilations of blood vessels that bear the risk of rupture and subsequent subarachnoidal haemorrhage, which is associated with high mortality and morbidity rates [1]. According to prevalence studies, around 3% of the world’s population hosts at least one of these aneurysms [2]. When identified, they raise the question of a potential clinical intervention, which also entails a non-negligible threat [3,4]. As a result, there is a need for risk evaluation tools in order to comprehensively assess the stability of IAs. So far, clinical decisions have mostly been based on the size, shape and location of the bulge. However, it was shown that the risk evaluation accuracy is limited when using only these parameters [5]. This is the reason why research effort is currently made to model and simulate patient-specific inner haemodynamics of IAs through computational fluid dynamics (CFD) in order to aid physicians in decision-making.

Computational models of aneurysm biomechanics indeed hold great promise for risk stratification, as haemodynamic features reveal key correlations with future aneurysm growth [6,7]. Performing numerical simulations for large numbers of aneurysm cases has raised multiple challenges in the scientific community, ranging from the efficient and systematic generation of adapted computational meshes to the solving of coupled systems of equations complemented with complex rheology models and tailored boundary conditions [8]. Most of these challenges aim at enriching the simulation fidelity towards real patient-specific predictive modelling. Among them, transitioning from rigid arterial wall modelling to adequate fluid–structure interaction (FSI) simulations stands as a key research goal that recently received attention through inspiring new contributions to aneurysm wall vibration analysis [9,10] and perianeurysmal contact constraints [11,12]. Pure CFD modelling based on the rigid wall assumption has been known for more than a decade to overestimate wall shear stresses (WSSs) [13,14], which, in turn, casts doubts in its ability to provide satisfactory criteria for rupture risk assessment. The fidelity could be refined by simulating the vascular flow in conjunction with vessel wall deformation via relevant coupled FSI modelling. The early research effort in this field was carried out by Torii et al. [13,15,16]. Authors compared the results of fully coupled FSI simulations using elastic and hyperelastic neo-Hookean wall behaviour [15]. Later, they investigated three aneurysms through elastic FSI simulations, comparing the obtained results with a fully rigid configuration [13]. It was suggested that the need for FSI modelling is geometry dependent, with WSSs in areas of flow impingement being overestimated in the rigid configuration. Going a step further, a few studies reported a specific interest in FSI simulations, where the wall thickness is modelled accurately [14,17]. Voß et al. imaged the geometry of a single aneurysm dome using micro-CT after the tissue was resected in surgical clipping [17]. Subsequent FSI simulations of the acquired geometry compared the results obtained with a uniform thickness configuration (0.3 mm) and with the specific thickness distribution measured in micro-CT, revealing peak local stress variations of around 50%. A few other research teams proposed similar studies [14,18,19] but always investigated different cohorts of aneurysm cases composed of a few specimens only. This, along with different modelling assumptions, undermines comparisons between them, thereby limiting the reproducibility of the reported results. At the same time, it is important to recall that the FSI in intracranial aneurysms is a complex multi-parametric problem, which would require large cohorts of investigated cases in order to draw robust conclusions. However, simulating fluid–structure coupled physics involves a certain algorithmic complexity and is computationally costly. In addition, until recently, clinical routine imaging techniques did not allow for the visualization of wall movement in the brain, thereby not encouraging the development of arterial compliance modelling for intracranial aneurysms. As a result, no general consensus has been reached on the relevance of FSI modelling in the context of IA risk assessment.

Over the past few years, IA pulsations were directly observed through clinical imaging [20,21,22,23], providing new perspectives to the FSI modelling of brain aneurysms. Indeed, in vivo data acquisition has substantially benefited the modelling of aortic aneurysms, with the prescription of patient-specific wall thicknesses [24] and even local tissue stiffness estimation through 4D flow analysis. However, this has hardly been conceivable in the brain, as most aneurysms have wall thicknesses ranging between 30 μm and 400 μm [25,26], hence falling under common medical imaging resolutions. Mostly supported by the development of very precise ECG-gated 4D-CTA [21,22,27], this situation is progressively changing and future research will surely benefit from additional in vivo data to feed FSI models. So far, aneurysm pulsation has mostly been expressed in terms of the overall bulge volume variation over a cardiac cycle, with peak reported values of 20% [23]. Even though these measurements suffer from large uncertainties, especially for small bulges [28,29], this new insight motivates the development of the FSI modelling of IAs beyond the scarce existing literature. Furthermore, publications have already reported that IAs demonstrate very different mechanical properties and thicknesses [26,30] due to several biological phenomena linked with their formation and growth [7], inducing potential local weaknesses in pathological tissue. If future medical equipment allows for the in vivo localization of these weaker spots, as was done post-mortem in [17], FSI models will surely contribute to building precise rupture risk assessment tools. On top of this, even if measurement data are still unavailable, assessing the sensitivity of several physical parameters in the context of FSI will give crucial insights for the future of IA modelling.

In this work, we introduce a new simulation setting using idealized geometries for the careful evaluation of different aneurysm dome shapes in interaction with pulsative blood flow. Although the analysis of patient-specific cases remains an ultimate goal, it seems that simplified IA geometries are still missing for studying FSI-related phenomena in a more controlled manner. Except for the work of Wan et al. [31], idealized geometries of sidewall aneurysms have always been investigated through rigid-wall CFD simulations [32,33,34], showing the sensitivity of haemodynamics towards various geometrical parameters. They were found to be very useful in particular for studying the impacts of certain modelling assumptions and boundary conditions [34], or highlighting specific and easily explainable trends that could be reproduced. In this work, we present novel FSI modelling based on the variational multiscale (VMS) method for both the fluid and solid solvers. This was employed to investigate the proposed benchmark setting, along with three real aneurysm domes. The introduced geometry is extremely versatile, as bulge shapes can be substituted while keeping the general case settings unchanged. This simplified problem offers a better environment to draw conclusions from a smaller manifold of explored configurations. It is idealized yet encompasses the important features of IA modelling so that the reported physical phenomena have the potential to be observed in real aneurysms. Furthermore, all geometric features, meshes, flow quantities, comparisons with the rigid wall model and corresponding plots are provided. We made sure that the test cases are easy to reproduce but help to draw the necessary conclusions regarding the system’s sensitivity to physical and geometrical parameters. The design of the proposed cases is described in the following section, along with the employed methods for solving the coupled physics. Then, we explore the possibilities of the proposed geometry and shed light on bulge shapes that might consequently benefit from the modelling of compliant arterial tissue.

## 2. Materials and Methods

### 2.1. Design of the Proposed FSI Case

#### 2.1.1. A Simple but Versatile Geometry

A cut view of the proposed geometry is shown in Figure 1. With an inlet diameter of 4 mm, the case mimicked a simplified sidewall aneurysm located on the internal carotid artery (ICA) [35]. This appears to be a relevant location, as more than half of all sidewall IAs are located around the ICA siphon or close to the ophthalmic artery branching [32,36]. The basic aneurysm bulge is designed as a perfect sphere intersecting the toroidal geometry. The proportions were inspired by previous work [33,34]. The wall thickness was set to 0.25 mm, which falls in the range of previously prescribed values in similar studies: from 0.2 mm [6,14] to 0.35 mm [18]. Given that this schematic is both unrealistic and a potential source of problems in finite-element simulations, the singular sharp angles at the neck were smoothed, resulting in the 3D geometry visible in Figure 2. In the following, we refer to this case as *R* (for Reference).

Going a step further, three realistic bulge shapes taken from [37] were employed by adapting them manually to fit the neck of our idealized geometry. The resulting shapes are presented in Figure 3 and are referred to as cases *A*, *B* and *C*.
Figure 2Overview of the case settings and parameters. The generalized inflow waveform was adapted from [38]. B.C. stands for boundary conditions.
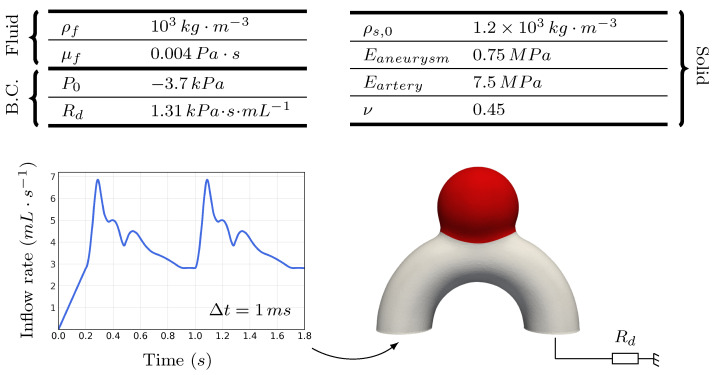

Figure 3Overview of the three cases adapted from the open-source *IntrA* dataset [37].
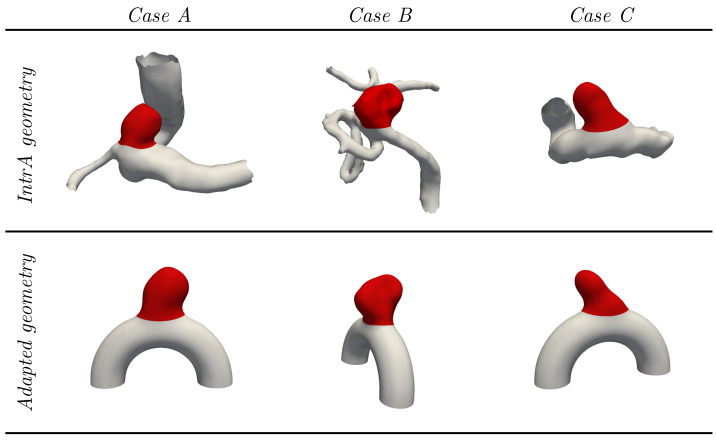


#### 2.1.2. Choice of Physical Parameters

All the employed physical parameters are summarized in Figure 2. As part of our idealized setting, the blood was modelled as a Newtonian fluid of viscosity $\mu_{f}$ = 0.004 Pa·s. For the walls, we followed previous FSI studies that modelled IAs, which mostly employed constant isotropic mechanical properties (E=1 MPa, ν=0.45) [14,15], with the neo-Hookean or Mooney–Rivlin models being commonly applied [13,14,15,17]. In our case, the arterial walls were ruled by neo-Hookean behaviour and we decided to draw the focus on the aneurysm bulge interaction with the flow by drastically raising the stiffness of the artery to $E_{artery}=10E_{aneurysm}$. We calibrated the bulge stiffness to $E_{aneurysm}=0.75$ MPa in order to reach aneurysm volume variations that are in line with recent wall pulsation in vivo observations [20,21,22,23]. The bulge area, where the lower stiffness was prescribed, corresponded to the part of the geometry that lay in the half-space y>8 mm.

#### 2.1.3. Boundary Conditions

In the fluid, inlet velocities and outlet pressure conditions were prescribed as follows:(1)$$\begin{array}{c} \begin{array}{cc} v(x,t) & =V\left(t\right)\left(1-{\left(\frac{||x-R_{torus}\left|\right|}{r_{torus}}\right)}^{2}\right)\;e_{y},\;x\in \Gamma_{f,in},\\ P(t) & =P_{0}+R_{d}\int_{\Gamma_{f,out}}v\left(x,t\right)\cdot \left(-e_{y}\right)\;d\Gamma ,\;\mathrm{on}\;\Gamma_{f,out}. \end{array} \end{array}$$
where V(t) was built based on the waveform plotted in Figure 2, corresponding to an averaged internal carotid pulse reported by [38]. It was prefixed with a 0.2 s linear ramp for a smoother initialization and scaled to reach an average flow rate of 4 mL·s^−1^, which correponded to the mean ICA flow measured among 13 studies, as reported in a recent review [39].

At the outflow, unlike in typical CFD simulations, the absolute value of the pressure was of major importance. To reach plausible deformations of the aneurysmal membrane, physiological pressures needed to be applied. Pressures in the vascular system range between 80 and 120 mmHg for healthy patients. These pressure variations occur over a cardiac cycle in the system and result from the hydraulic resistance of the posterior vasculature ($R_{d}$), and are mostly imputable to brain capillaries. To account for this, the outflow pressure was scaled with respect to the flow rate (i.e., adjusting $R_{d}$) to keep the pressure in the system between the given bounds, similar to [15]. As part of our idealized setting, we decided to vary the outflow pressure between 0 and 40 mmHg (5.3 kPa) rather than pre-stressing the diastolic structure, as was done by [14]. Thus, we set $P_{0}=-3.7$ kPa and $R_{d}=1.31$ kPa·s·mL^−1^. For the solid, nodes situated on the inflow/outflow plane (y=0) were kept fixed, whereas a traction-free condition was prescribed on $\Gamma_{s,ext}$.

#### 2.1.4. Quantities of Interest

One of the major goals of the proposed practical case lay in the ability to assess the sensitivity of haemodynamics to FSI modelling. The most widely used metric for the rupture risk assessment of IAs is the wall shear stress (WSS) applied by the blood flow in the bulge. The WSS is associated with remodelling pathways of IAs [6,7,40,41], leading to their formation and growth due to the interaction between abnormal blood flow and the endothelial cells of the vessels [42]. As multiple definitions exist, the one employed in this work is provided in Equation (Equation 2) for the sake of reproducibility:(2)$$\begin{array}{c} \tau_{WSS}=n\times \left[\left(\sigma_{f}\cdot n\right)\times n\right]=\sigma_{f}\cdot n-\left[\left(\sigma_{f}\cdot n\right)\cdot n\right]n \end{array}$$
where n is the unit normal vector at the wall and $\sigma_{f}$ is the Cauchy stress tensor defined as $\sigma_{f}=-p_{f}I+\mu_{f}\left(\nabla v+\nabla^{T}v\right)$. This vectorial definition allows for computing another important metric known as the oscillatory shear index (OSI):(3)$$\mathrm{OSI}=\frac{1}{2}\left(1-\frac{\left|\right|\int_{t_{0}}^{t_{0}+T}\tau_{WSS}\;dt\left|\right|}{\int_{t_{0}}^{t_{0}+T}\left|\right|\tau_{WSS}\left|\right|\;dt}\right)$$

In the presented results, we emphasize these two indicators and record them over the second cardiac cycle only (from $t_{0}=1$ s to $t_{0}+T=1.8$ s) in order to limit any transient effect associated with the initial flow development. The WSS norm was averaged over this cardiac cycle to yield the time-averaged WSS (TAWSS).

#### 2.1.5. Meshing

The mesh resolution is of major importance, especially in the vicinity of the walls, to properly resolve velocity gradients. Therefore, we used a boundary layer in the fluid domain, as shown in Figure 4. We employed a geometrical progression (factor α=1.2) between successive layers, a minimal element thickness of 0.02 mm and a total boundary layer size of 0.3 mm. The isotropic element size of the core mesh was set to 0.17 mm, and the solid thickness was divided into 6 equal layers of 0.042 mm. This resulted in a mesh composed of 1.03 M and 0.56 M elements for the fluid and the solid of the reference case (*R*), respectively. All the meshes were generated with the Gmsh [43] Python package and are available on GitHub (https://github.com/aurelegoetz/AnXplore accessed on the 8 March 2024).

### 2.2. Modelling the Physics

The employed FSI implementation was based on our C++ in-house library. It is a highly parallelized stabilized finite-element code that relies on PETSc. Both our fluid and solid mathematical formulations are described in detail in the following sections. Due to the high modularity of FSI solvers, each framework was unique and needed to be tested on simple benchmarks. The employed method was validated using the well-known Turek benchmark [44] and the pressure wave case proposed in [45], which appears to be the only widespread reference for FSI haemodynamics. The later validation is reported in the Appendix, along with convergence studies in time and space for the proposed case *R*.

#### 2.2.1. Fluid Mechanics

When simulating haemodynamics in compliant arteries, keeping a fitted fluid–structure coupling interface is decisive in obtaining precise WSS estimates (see Section 2.1.4). Furthermore, as mesh deformations remain moderate, an arbitrary Lagrangian–Eulerian (ALE) [46] description appears the most practical approach to employ. Thus, we defined $\Omega_{f,t}\subset \;{I\;R}^{n}$ as the fluid spatial domain at time t∈[0,T], with *n* as the spatial dimension and ψ as the ALE mapping from $\Omega_{f,0}$ to $\Omega_{f,t}$. The associated relative velocity is denoted $v_{m}$. Let $\Gamma_{f}$ be the boundary of $\Omega_{f}$. We considered the mixed formulation in velocity v and pressure $p_{f}$ of the transient incompressible Navier–Stokes equations given by
(4)$$\rho_{f}\partial_{t}v+\rho_{f}\left(\left(v-v_{m}\right)\cdot \nabla \right)\;v-\nabla \cdot \sigma_{f}=f,\;\mathrm{in}\;\Omega_{f,t}.$$
(5)$$\nabla \cdot v=0,\;\mathrm{in}\;\Omega_{f,t}.$$
where $\rho_{f}$ is the fluid density and f is the source term. The time derivative is implemented with a second-order backward differentiation formula (BDF). The mixed formulation is based on a P1-P1 discretization, combined with a variational multiscale (VMS) method, as described in [47]. The VMS method ensures accuracy and stability [48], even for convection-dominated flow by enriching both velocity and pressure with residual-based subscales.

In the ALE framework, the convective velocity is altered by the mesh velocity $v_{m}$, which tracks the movement of coupling interfaces ($\Gamma_{FSI}$). The adaptive mesh displacement allows for keeping boundaries fit and consequently saves the cost of interpolating between subdomains. A $C_{2}$-smooth $v_{m}$ field can be obtained, for instance, by solving the following diffusion equation [49]:(6)$$\begin{array}{c} \begin{array}{cc} \nabla \cdot (\gamma \nabla v_{m}) & =0,\;\mathrm{on}\;\Omega_{f},\\ v_{m} & =\partial_{t}u,\;\mathrm{on}\;\Gamma_{FSI},\\ v_{m} & =0,\;\mathrm{on}\;\Gamma_{f}\setminus \Gamma_{FSI}, \end{array} \end{array}$$
where u stands for the displacement of the solid interacting with the considered fluid domain. The diffusion coefficient γ was taken to be the squared inverse distance to the interface $\Gamma_{FSI}$ in order to better share the mesh deformation on the entire grid and to keep the boundary layer mesh as intact as possible. Many other solutions exist and the interested reader can refer to [50,51].

#### 2.2.2. Solid Mechanics

The compliant arterial tissue was modelled using the Lagrangian equations of solid dynamics. Let $\Omega_{s,0}$ and $\Omega_{s,t}$ define the initial and current solid spatial domains, with ϕ being the mapping between the two domains. We should distinguish here between the material Lagrangian coordinate X and the updated Lagrangian coordinate x. The displacement of a solid particle is given by u=x−X=ϕ(X,t)−X and the deformation gradient is defined as $F=\nabla_{X}\phi$. The Jacobian determinant is thus J=det[F]. The momentum and continuity equations for the solid dynamics were specified as follows:(7)$$\rho_{s}\partial_{tt}u-\nabla_{x}\cdot \;\sigma_{s}=0,\;\mathrm{in}\;\Omega_{s,t}.$$
(8)$$\rho_{s}J=\rho_{s_{0}},\;\mathrm{in}\;\Omega_{s,t}.$$
where $\rho_{s}$, $\ddot{u}$ and $\sigma_{s}$ designate the solid density, the second material derivative of the displacement and the symmetric Cauchy stress tensor, respectively. As for the fluid, BDF2 was employed for the acceleration term.

To model the intrinsically hyperelastic nature of arterial tissue [14,15,52], we relied on the Helmholtz free energy formalism. Let C denote the right Cauchy–Green strain tensor given by $C=F^{T}F$ and $S=JF^{-1}\sigma_{s}F^{-T}$ denote the second Piola–Kirchhoff stress tensor. The Helmholtz free energy function Ψ(C) is defined by
(9)$$S=2\partial_{C}\Psi \left(C\right).$$

This free energy function is decomposed into its volumetric and deviatoric contributions, leading to the classical split:(10)$$\Psi \left(C\right)=U\left(J\right)+W\left(\bar{C}\right).$$
where $J=\sqrt{\det[C]}$, and $\bar{C}=J^{-\frac{2}{3}}C$ is the volumetric/deviatoric part of C.

We considered a neo-Hookean and Simo–Taylor [53] volumetric model, which yields
(11)$$U\left(J\right)=\frac{1}{4}\kappa \left(J^{2}-1\right)-\frac{1}{2}\kappa \ln J,$$
(12)$$W\left(\bar{C}\right)=\frac{1}{2}\mu_{s}\left(\mathrm{tr}\left[\bar{C}\right]-3\right)=\frac{1}{2}\mu_{s}\left(\bar{I_{1}}-3\right).$$
where κ and $\mu_{s}$ are material properties, and $I_{1}=\mathrm{tr}\left[\bar{C}\right]$ is the first Cauchy–Green invariant. The Cauchy stress tensor can similarly be split into its deviatoric and volumetric parts, which gives
(13)$$\sigma_{s}=p_{s}I+\mathrm{dev}\left[\sigma_{s}\right].$$
(14)$$p_{s}=2J^{-1}F\partial_{C}U\left(J\right)F^{T}=U^{\prime }\left(J\right)=\frac{1}{2}\kappa \left(J+J^{-1}\right),$$
(15)$$\mathrm{dev}\left[\sigma_{s}\right]=2J^{-1}F\partial_{C}W\left(\bar{C}\right)F^{T}=\mu_{s}J^{-\frac{5}{3}}\mathrm{dev}\left[{\mathrm{FF}}^{T}\right].$$

The final system of equations to be solved is given by
(16)$$\rho_{s}\partial_{tt}u-\nabla_{x}p_{s}-\nabla_{x}\cdot \;\mathrm{dev}\left[\sigma_{s}\right]=0,\;\mathrm{in}\;\Omega_{s}.$$
(17)$$\nabla_{x}\cdot u-\frac{1}{\kappa }p_{s}=0,\;\mathrm{in}\;\Omega_{s}.$$

Similar to the fluid, a VMS method was employed. More details about the implementation are given in [54].

#### 2.2.3. Coupling

Dynamic and kinematic coupling conditions (Equation 18) must be enforced at the fluid–solid interface, whose normal field is denoted n:(18)$$\begin{array}{c} \begin{array}{cc} v & =\partial_{t}u,\;\mathrm{on}\;\Gamma_{FSI},\\ \sigma_{f}\cdot n & =\sigma_{s}\cdot n,\;\mathrm{on}\;\Gamma_{FSI}. \end{array} \end{array}$$

The interface continuity constraints were imposed using a partitioned, iterative scheme [55] with classic Dirichlet-to-Neumann coupling. This sub-iterative process appears especially crucial when fluid and solid densities come close or when dealing with slender solid geometries in order to not suffer from coupling instabilities commonly known as the added-mass effect [56,57].

Reaching FSI convergence can be mathematically viewed as finding a fixed point of the solid and fluid operators’ composition (S∘F). We define the FSI residual as follows:(19)$$r_{t}^{k}=u_{t}^{k}-{\tilde{u}}_{t}^{k}=S\circ F\left({\tilde{u}}_{t}^{k}\right)-{\tilde{u}}_{t}^{k},$$
where ${\tilde{u}}_{t}^{k}$ is the predicted displacement of the solid used at sub-increment *k*. Time was only incremented after this fixed point was reached with a given tolerance $\left|\right|r_{t}^{k}\left|\right|<tol_{FSI}$ (||.|| is the Euclidian norm scaled with the number of nodes in the mesh). For the proposed test cases, this tolerance was set to $10^{-5}$ mm. For the fixed-point algorithm to converge quicker, under-relaxation is widely employed [49,58,59]. This consists of using only a fraction of the algorithm’s new solution ($u_{t}^{k}$) for building the next guess. Mathematically, relaxing the solution with a relaxation parameter ω can be written as
(20)$${\tilde{u}}_{t}^{k+1}={\tilde{u}}_{t}^{k}+\omega \left(u_{t}^{k}-{\tilde{u}}_{t}^{k}\right)={\tilde{u}}_{t}^{k}+\omega r_{t}^{k}$$

From within the large family of relaxation methods [60], we chose the momentum-accelerated Aitken Δ2 scheme due to its well-studied properties and extensive use in the community. The dynamic relaxation parameter $\omega_{t}^{k}$ can be assessed at every sub-increment *k* by using the following formula [60]:(21)$$\omega_{t}^{k}=-\omega_{t}^{k-1}\frac{{\left(r_{t}^{k-1}\right)}^{T}\left(r_{t}^{k}-r_{t}^{k-1}\right)}{\left|\right|\left(r_{t}^{k}-r_{t}^{k-1}\right){\left|\right|}^{2}}$$

However, one should note that Equation (Equation 20) cannot be used to initialize the predicted displacement ${\tilde{u}}_{t}^{1}$ when starting a new time step, as no previous sub-increment exists. As a result, a linear predictor was employed based on the previous converged displacement. Similarly, Equation (Equation 21) could only be used from the end of the second sub-increment. Before that, a fixed value $\omega_{0}=0.1$ was employed.

## 3. Results

### 3.1. FSI in the Proposed Idealized Aneurysm

Before moving to complex bulge shapes, the impact of arterial wall modelling on the spherical aneurysm is assessed in this section. The employed meshes and time step (Δt=1 ms) were selected based on time and space convergence studies reported in the Appendix. The computations were carried out on dual processors (32-Core AMD EPYC 64-bit Processor 7502, Advanced Micro Devices, Santa Clara, CA, USA) with a 2.5 GHz base clock rate and HDR 100 interconnection. With this hardware, simulating 1.8 s of physical time took 2 h and 17 h for the rigid and deformable wall settings, respectively.

Figure 5 details the obtained systolic (t=1.093 s) haemodynamics for the deformable case, whose Reynolds and Womersley numbers peaked at 520 and 2.8 in the parent vessel. The inlet parabolic blood flow was rapidly altered by the vessel’s curvature and separated at the neck, with a small fraction entering the bulge, impinging on the aneurysm wall, and resulting in high WSS values (TAWSS rose up to 15.7 Pa in the bulge). The impinging jet later recircultated and exited the aneurysm at the sides. We measured the intensity of this swirl by recording the upward-going flow rate through the horizontal plane crossing the aneurysm centre at y=10 mm. The swirl took time to fully develop, as illustrated by the 16 ms delay between its maximum intensity and the maximum inflow rate shown in Figure 5b. Interestingly, this time shift was almost doubled to 26 ms when negelecting the wall compliance, likely due to the absence of the wall’s sucking effect associated with the aneurysm pressure-induced inflation. Apart from its earlier development, the bulge’s inner recirculation also extended over a wider part of the wall in the compliant configuration. Indeed, due to the opening of the aneurysm neck in the *z*-direction shown by the displacement field in Figure 5, the flow spread over larger sections, which also resulted in lower velocity levels compared with the rigid counterpart. At the horizontal section crossing the middle of the aneurysm bulge, the peak velocities decreased by 3% when modelling the deformable walls compared with a fully rigid resolution. At the neck, the systolic velocities decreased by around 20%, as shown in Figure 5a. It can also be seen on this plot that employing deformable walls slightly shifted the impinging flow to the distal side of the neck compared with the rigid case due to its wider opening. This region highlighted at the neck appeared to be the most sensitive to changes in the mesh size and time step. Thus, the proposed line plot (Figure 5a) of the vertical velocity component $v_{y}$ could serve as a reference for reproducing the case and was used in the convergence study (see Appendix B). The curves were given for both rigid and compliant models to ease a step-by-step reproduction of the presented results.

As for the pressure, Figure 5d reveals a time delay of τ=4 ms between the inflow rate peak and the maximum outflow pressure. The tissue compliance allowed for delayed and damped flow variations between the inlet and outlet, which directly affected the prescribed boundary conditions (Equation 1). In terms of spatial distribution, pressure values continuously decreased along the vessel’s centreline with concentration spots on the outer curve of the inflow vessel and in the impingement area at the neck. This pressure field dominantly dictated the wall’s expansion movement over shear forces, resulting in almost no tilting of the aneurysm in response to the inflow jet at the neck, agreeing with previous observations [15]. The recorded volume variation and displacements (see Figure 6) lay in the range of previously reported values [20,23].

The impact of the modelling choice on the quantities of interest (see Section 2.1.4) is reported in Figure 7 and Figure 8. As the bulge opened under fluid stress, the wall inclination at the impingement area changed, causing the fluid jet to sweep over a larger part of the neck, with a slightly more horizontal jet. This explained the alternation of higher and lower TAWSS for the FSI modelling in Figure 7. Apart from this specific spot, compliant wall modelling generally lowered the WSS values in the bulge for this case. This is also highlighted in Figure 5c, where the WSS is followed over time. There, the norm of $\tau_{WSS}$ (see Equation (Equation 2)) was integrated over the surface of the bulge to obtain a WSS spatial average over this region of interest. Observed values were consistently lower in the compliant wall configuration, with the peak values dropping by 8.5%. This confirmed the trends observed in previous studies [13,14], although the magnitude of the changes was smaller in our idealized sidewall geometry for which the intra-saccular flow was mostly shear-driven, as opposed to the strong impingement configurations observed in some bifurcation aneurysms. It is also interesting to note the slight time shift between the rigid and deformable curves in Figure 5c that can be linked with the delayed initiation of the aneurysmal swirl in the rigid case, as was already pinpointed using Figure 5b. Regarding the OSI, the computed values were barely altered, as shown in Figure 8. Only a slight shift of the high values towards the back of the bulge was observed due to the deeper flow penetration into the aneurysm dome. If the embedded symmetry of this reference case (*R*) is naturally reflected in flow patterns and results in subtle compliance-driven changes, more complex geometries reveal more drastic alterations, as shown in the following section.

### 3.2. Exploring Different Bulge Shapes

To explore implications beyond a fully spherical bulge, the specific cases *A*, *B* and *C* are thoroughly examined in this section. Please note that by construction, all the cases had the same neck geometry, which allowed for drawing the focus solely on the bulge’s topology.

The three cases shared some general flow aspects with the spherical geometry. Using FSI modelling, the flow swirled deeper inside the bulge, as illustrated by the shift in the OSI values (see Figure 8) and the impingement area swept over a larger part of the neck, smearing out the WSS pattern there. Overall, the TAWSS values decreased on the aneurysm wall opposed to the flow, while they consistently rose close to the dome, where the wall movements imposed the slow flow dynamics.

Although inspired by a real aneurysm shape, case *A* did not deviate significantly from the reference case, even when looking at the details. Cases *B* and *C*, per contra, had very different flow mechanisms, leading to large deviations between rigid and compliant tissue modelling. First, case *B* featured a weak structural spot, which consequently deformed under internal pressure due to its negative curvature, as displayed in Figure 6. The induced change in geometry reoriented the inflow jet and locally reduced by more than 15% of the TAWSS values at the aneurysm distal wall. As indicated by the dotted annotations in Figure 9, the initially split aneurysmal flow (rigid configuration) was merged in one general swirl for the compliant configuration due to the outward bulging of the solid membrane. As a result, the dome’s exposure to the flow increased, as illustrated by the consequent drop in the OSI in that area displayed in Figure 8, which revealed the alteration of the almost stagnating flow that was observed there in the rigid case. In this patch, TAWSS also increased fivefold when moving from rigid to compliant wall modelling, stressing the major change in the local flow pattern.

The last investigated case, namely, case *C*, also showed a noticeable change in the flow pattern when employing compliant modelling. Contrary to the other two, the flow did not easily penetrate the dome due to the orientation of the bulge with respect to the parent vessel. This resulted in a secondary recirculation in the dome in the rigid case, which is commonly observed in high-aspect-ratio IAs. This slow recirculation was significantly altered when compliant modelling was employed due to its low inertia. The rotation of the flow was indeed modified by the expansion and contraction of the wall, as emphasized by the dotted annotations in Figure 9. The OSI pattern was thus largely impacted, as it was very sensitive to flow orientation (see Figure 8). Even though the general TAWSS distribution was not altered significantly in that case, the low TAWSS values around the aneurysm dome still locally increased by more than 50%, indicating the enhanced flow dynamics there. Overall, the reported results show how different bulge shapes may demonstrate various degrees of responses to compliant tissue modelling.

## 4. Discussion

### 4.1. Impact of the Wall Modelling

The initiation and growth of IAs are caused by abnormal blood flow conditions, which can be characteristized by haemodynamic metrics, such as the ones investigated in this work [7]. The most commonly studied remains the WSS, for which both high and low values are considered a threat to an IA’s stability over time. If high values emphasize areas of fluid stress concentration, the lower ones are associated with blood stagnation, which can trigger an inflammatory response of the walls and atherosclerosis development [61]. In this work, we witnessed a larger spreading of impinging fluid jets when compliant walls were employed, resulting in an overall smaller range of encountered TAWSS values in the bulge. In particular, the low TAWSS values close to the aneurysm dome have been reported to substantially increase, as wall movement helps to initiate flow dynamics in stagnating regions. However, one should note that WSS values are often compared with given thresholds for risk assessment [42], which potentially mitigates the importance of that change in the minimum values if they remain under classically employed thresholds. This WSS alteration has already been reported by previous studies [14,15], but less attention has been drawn to the oscillatory flow behaviour when it comes to FSI modelling.

The OSI complements the description of haemodynamics at the wall’s vicinity by giving insights on flow orientation changes over a cardiac cycle. Logically, flow inversions mostly occur in stagnation areas, which are likely to be affected by wall movement. When the WSS values varied by a few percent when considering wall compliance, the OSI distributions were shown to drastically vary with the kind of wall treatment. This is illustrated by Figure 8, where substantial OSI variations were observed for cases *B* and *C*. For the latter case, the general haemodynamics changed at the dome, moving from a steady swirl to a contraction–expansion motion following the wall movement. The OSI naturally reflects these changes at the surface and reveals the greater interest of compliant wall modelling. The fact that this metric can change significantly depending on the employed wall modelling suggests that rigid CFD results may yield inaccurate risk estimation. A comparable conclusion has been reported in a recent study, where the authors show the impact of wall modelling on the gradient oscillatory number [62], which is another metric for characteristizing oscillatory flow. As some thrombus models are built based on flow oscillatory indices [63], the pinpointed changes appear relevant in terms of clot formation risk assessment. Even though not studied here, the flow residence time is also a commonly employed indicator to predict potential thrombus formation for numerically evaluating the outcome of flow diverters [64]. Such a flow metric will also undoubtedly reveal a different behaviour using compliant modelling and should be explored in future work to reveal the impact on different aneurysms. All in all, considering that bulge geometries with daughter sacs or intricate dome topologies are very common and often feature zones of blood stagnation, compliant wall modelling may be necessary to retrieve accurate slow flow patterns.

While cases *B* and *C* featured some noticeable differences in terms of the classical haemodynamic risk metrics, the results show that case *A*’s haemodynamics were barely altered, which was certainly because of the case’s regular shape and the bulge alignment with the impinging flow. The same was true for the reference case, which did not manifest any major haemodynamic alteration due to wall modelling, but gave a common ground for reproducibility in this study and helped with stressing the general flow alteration mechanisms common to all cases. The diverse responses towards wall modelling observed in this study led to the hypothesis that the compliance-related effects were strongly shape dependent. This motivated our sensitivity study, along with future research effort to assess the generalizability of the drawn conclusions onto large-scale patient cohorts and to help identify critical cases for which compliant modelling would benefit the most. Contrary to other patient-specific data inputs, the lumen’s topology is systemically acquired in clinical routine when detecting an aneurysm, already rendering such studies feasible. However, as our results suggest, particular attention has to be dedicated to the segmentation quality of magnetic resonance angiography, along with the level of smoothing employed when generating the computational domains [65], as topological details can yield large deviations when compliant modelling is employed.

Lastly, an important aspect regarding outflow boundary conditions has to be stressed. It is interesting to note that almost all previous publications reporting FSI simulations of IAs investigated bifurcation cases [14,15,17,18,25]. In this study, a single outlet branch was considered for the sake of simplicity, thus circumventing the treatment of the outflow split. If outflow boundary conditions were already shown to be a sensitive aspect of CFD simulations [34], they are even more so for FSI. Indeed, several outflow branches featuring different radii will inevitably expand differently under internal pressure, thus altering the hydraulic resistance of the outflow paths and modifying the flow split. This change surely impacts the general flow pattern consequently. This observation suggests that the reported differences between rigid-wall and FSI simulations could be amplified when considering intricate patient-specific geometries, which almost always feature vascular branches in the vicinity of the aneurysm.

### 4.2. Limitations and Perspectives

In the design of our idealized setting, we targeted a balance between simplicity and completeness to provide test cases that are both easy to reproduce and close enough to real aneurysm physics. In order to bridge the gap with patient-specific high-fidelity simulations, there are three main aspects that could be extended. First, blood should be modelled as a non-Newtonian fluid using patient-tailored rheology [66]. Recent advances have been made in modelling the behaviour of blood [67,68], providing complex thixotropic models that come closer to reality in the aim of assessing patient cases and that could potentially alter the computed haemodynamic metrics. Second, arterial wall modelling could be improved by both pre-loading the structure in its diastolic state, as was done in [14], and by enriching the employed behaviour law. Lowering pressure levels instead of pre-stressing the solid involves a different region of the stress–strain curve, leading to an overestimation of the reached deformations for strain-stiffening materials [69]. In this work, the bias was limited by calibrating mechanical properties to reach plausible levels of deformation, but spatial distributions may vary when considering pre-stress. However, this effect appeared secondary as part of the proposed idealized setting and the employed approach facilitated the reproduction of the presented results. Concerning the behaviour law, picking a more elaborate modelling method is currently undermined by uncertainty regarding aneurysmal tissue characteristics. Research effort was carried out to measure the material properties of resected IA domes after a clipping surgical operation [30,70,71], analyze the alteration of collagen fibre architectures, and report a substantial scatter in ultimate stress and Young moduli between specimens. Even on the surface of a single bulge, it was shown that distinct regions can exhibit very different material properties [71,72]. Although implementing more realistic anisotropic models, such as HGO [52], does not represent a technical challenge, applying them in a meaningful way is arduous, as the literature does not provide insights into the preferred orientation of fibres in pathological tissue. This has comforted us in the choice of a neo-Hookean model, but it could be replaced in the future by a Mooney–Rivlin model, as recent work underlined its better fit for the application at hand [73]. Taking a further step, the viscoelastic nature of arterial walls [74] could be taken into account, as tissue relaxation may alter the system’s coupled dynamics.

The last main aspect that creates a gap with patient-specific cases lies in the proposed case geometry itself. Having a symmetric parent vessel and inlet profile limits the complexity of the developing aneurysmal flow compared with real tortuous arteries. Non-symmetrical flow profiles featuring a misalignment with the parent vessel’s centreline could be employed to improve the completeness of the case. Overall, the relevance of these modelling details and their impact on the drawn conclusions in such a simplified setting remain unclear, motivating our inclination to simplicity.

We hope that progress in imaging technology will give more information about missing physical parameters, allowing accurate FSI modelling tools to demonstrate their efficiency. If locally varying wall properties can be assessed in vivo, FSI simulations will appear even more relevant. The in vivo observation of aneurysm pulsations represents a first step, as inverse analysis could help with tuning tissue mechanical models for patient-specific aneurysms. As a local tissue weakness constitutes a critical stress concentration point that jeopardize an IA’s stability, FSI will surely contribute to high-fidelity risk assessment tools. Meanwhile, validating the reported trends on large-scale cohorts remains one of our priorities, as previous studies remained limited to a few investigated cases. While only four shapes were considered as part of this study, the versatility of the proposed case allows us to explore a large manifold of realistic aneurysms in a controlled environment. Underlining aneurysm phenotypes that mostly benefit from the compliant modelling of arterial tissue stands as one of our future research goals.

## 5. Conclusions

This work introduced a novel idealized sidewall aneurysm geometry for assessing the relevance of FSI modelling in several easily reproducible configurations. Specifically, four aneurysm bulge shapes were investigated using the proposed framework, keeping all the other simulation parameters unchanged. Well-known haemodynamic metrics, such as the WSS and OSI, were computed using both rigid and compliant wall modelling, revealing significant flow changes linked with the aneurysm topology. Our conclusions show that modelling compliant vessels both altered the computed WSS values, as previously reported, and restructured the observed flow patterns, especially in slow blood recirculations, where the OSI could be drastically modified. Furthermore, the sensitivity to the employed wall modelling was shown to strongly depend on the aneurysm bulge shape. Differences between rigid and compliant modelling could even be amplified using patient-specific pathological tissue data, although the literature does not provide insights yet. Progress in medical imaging, along with large-scale studies, will certainly help to stress the limits of the widely employed rigid wall assumption. In all scenarios, FSI models, like the one introduced in this work, have to be developed to improve the comprehension of IAs and to assess the relevance of wall tissue modelling assumptions.

## Figures and Tables

**Figure 1 bioengineering-11-00269-f001:**
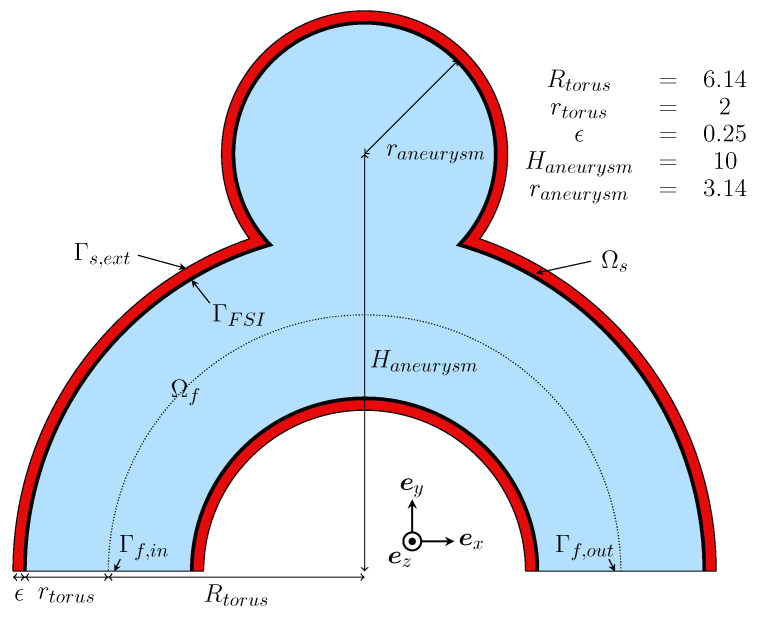
Schematic illustration of the proposed case *R*. Dimensions are given in mm.

**Figure 4 bioengineering-11-00269-f004:**
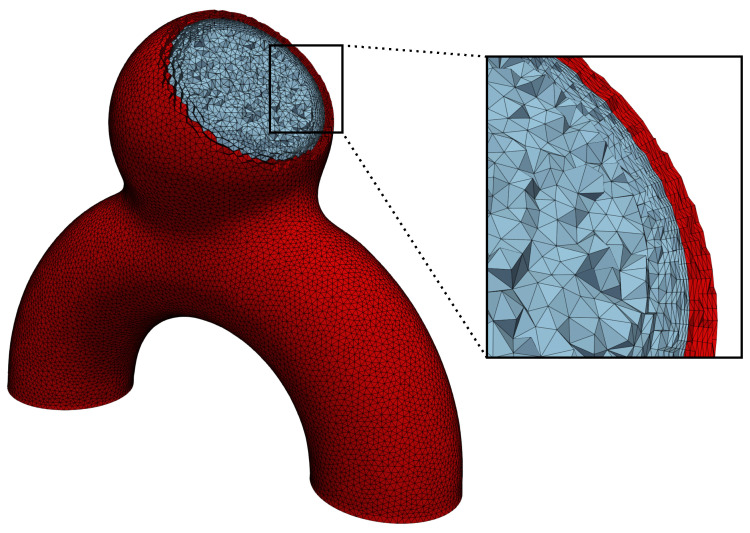
Views of the employed meshes for case *R*. The zoomed-in window on the right emphasizes the boundary layer at the vicinity of the walls.

**Figure 5 bioengineering-11-00269-f005:**
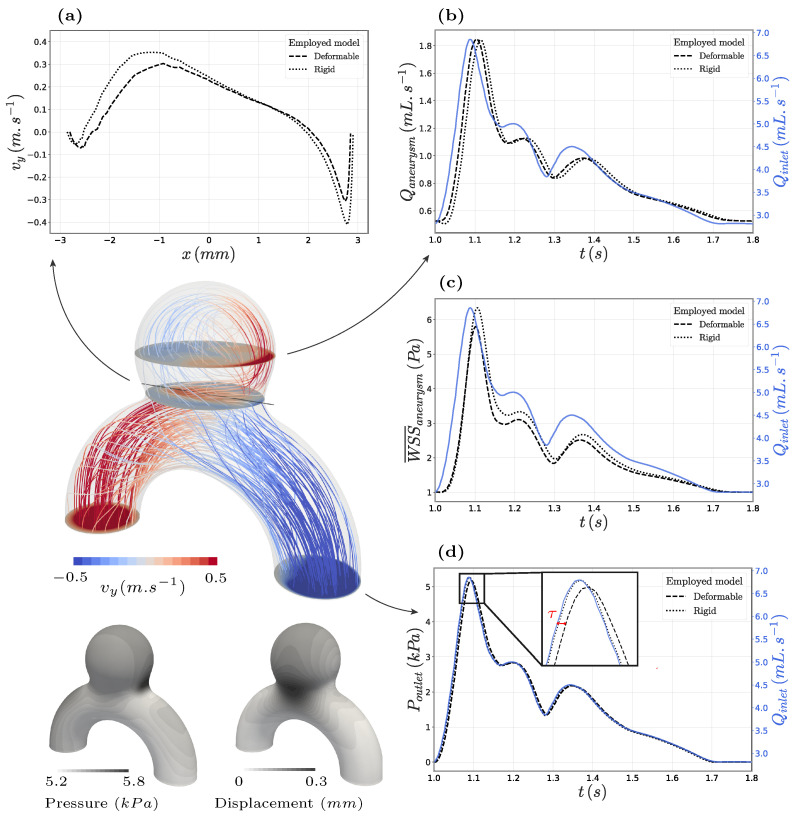
Detailed flow analysis of the proposed FSI case *R*. Systolic flow lines and pressure and surface displacement contours are displayed on the left side of the panel. They are complemented by four graphs, which detail the observed haemodynamics and compare them with the equivalent rigid configuration. The parent vessel’s inflow rate is overlaid on all three graphs on the right for visualizing time shifts over the cardiac cycle. (**a**) First, the velocity vertical component is plotted along the grey line traversing the neck as a reference curve. (**b**) On the right, the upward-going flow rate traversing the aneurysm bulge at its centre’s altitude (y=10 mm) is displayed. (**c**) Below, we report the surface-averaged WSS over the aneurysm bulge (defined by y>8 mm). (**d**) Lastly, we depict the pressure variations at the system’s outlet.

**Figure 6 bioengineering-11-00269-f006:**
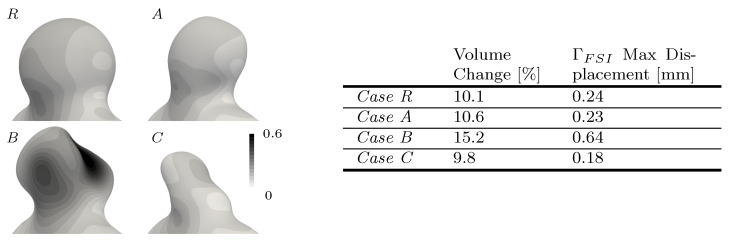
Systolic displacement field for the investigated cases. Displacements of the FSI coupling interface are displayed on the left (in mm). Maximum aneurysm volume variation and displacements are reported in the table on the right. The aneurysm volume variation was computed with respect to the diastolic state by integrating the volume situated over the plane y=8 mm.

**Figure 7 bioengineering-11-00269-f007:**
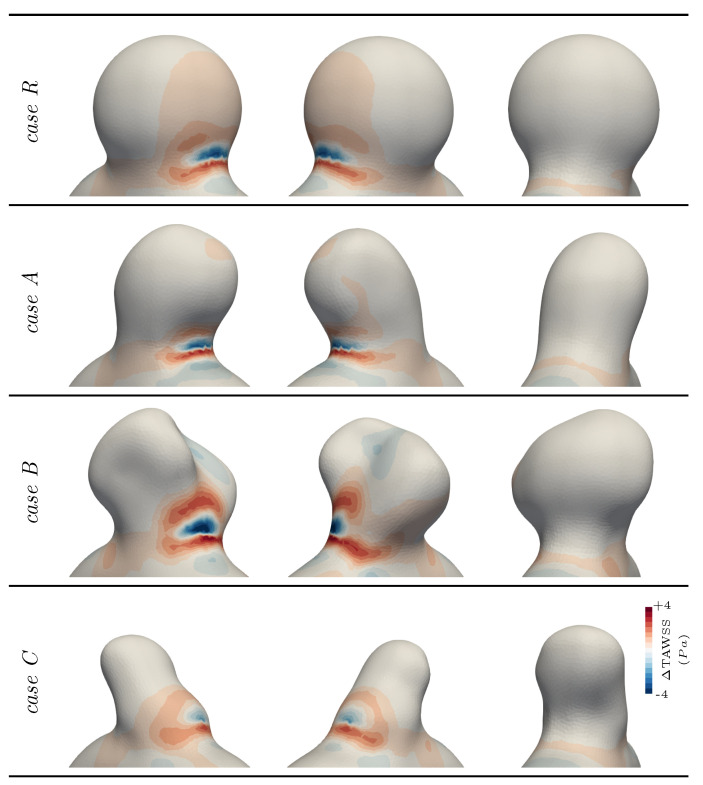
TAWSS differences between the two wall modelling assumptions (rigid − compliant).

**Figure 8 bioengineering-11-00269-f008:**
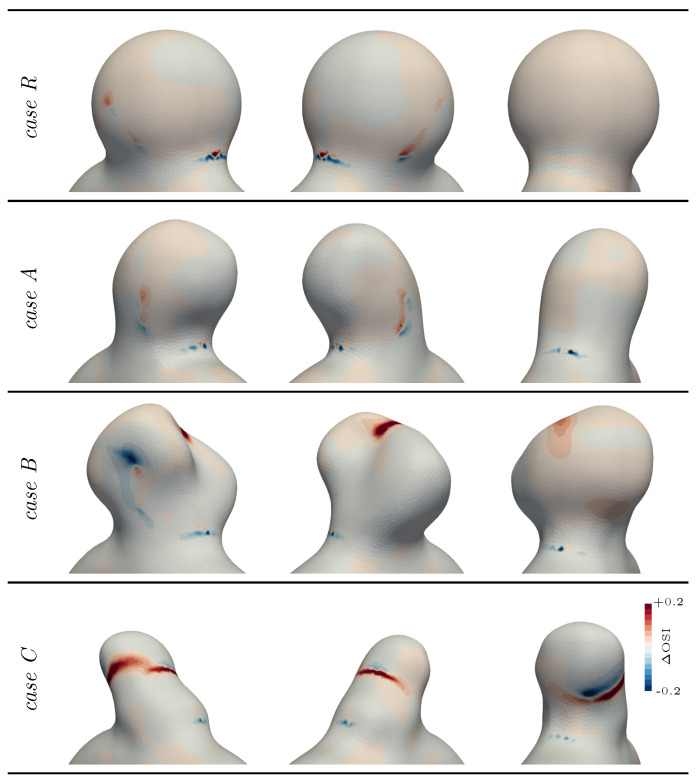
OSI differences between the two wall modelling assumptions (rigid − compliant).

**Figure 9 bioengineering-11-00269-f009:**
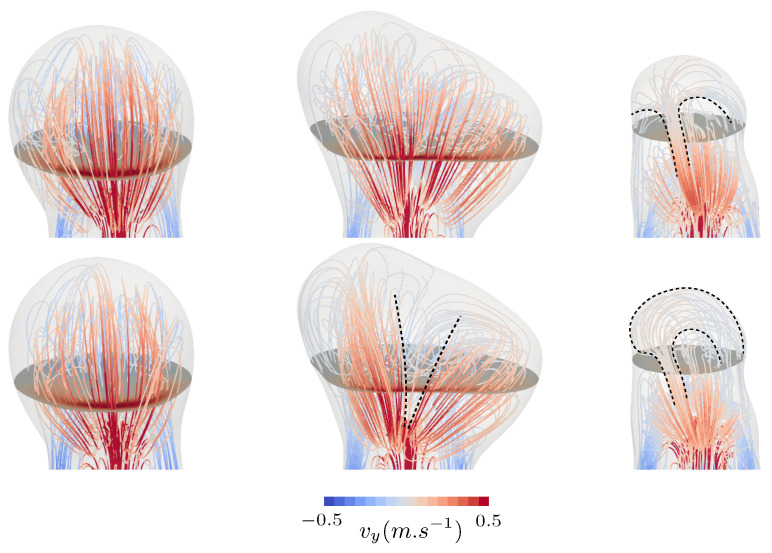
Systolic flowlines for the three specific bulge shapes. Velocity streamlines are shown for cases *A*, *B* and *C* to illustrate the shape of the flow inside the aneurysm bulge using compliant wall modelling (**top** row) compared with a fully rigid treatment (**bottom** row).

## Data Availability

The computational meshes of the cases investigated in this study are openly available on GitHub https://github.com/aurelegoetz/AnXplore (accessed on 30 January 2024).

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
