# Peer review of "Analysis of Intracranial Aneurysm Haemodynamics Altered by Wall Movement"

_bioengineering, 2024, doi:10.3390/bioengineering11030269_

Round 1

Reviewer 1 Report

Comments and Suggestions for Authors

In general, this is a good paper that studies blood flow in three typical anurisms using moving walls (FSI) fromultaion. The authors use the Variational Multiscale Method, which they have implemented and tested themselves using well-known benchmarks. The text is well designed and well written. Sufficient details of previous works, current work, limitations and perspectives are given.

Author Response

Dear reviewer,

Thank you very much for your positive feedback.

We are pleased to hear that you found this publication to be well designed and well written.

Best regards.

Reviewer 2 Report

Comments and Suggestions for Authors
  1. 1. The paper appears to lack significant novelty in the field, resembling more of a technical report than a scientific contribution with groundbreaking findings.

  2. 2. The authors would benefit from considering the research conducted by the Karniadakis group, particularly their exploration of hemodynamics in aneurysmal geometries and the associated instabilities. Relevant works include H. Baek et al.'s study on flow instability and wall shear stress variation in intracranial aneurysms (Journal of the Royal Society Interface, 2010) and Y. Yu et al.'s work on fractional modeling of viscoelasticity in 3D cerebral arteries and aneurysms (Journal of Computational Physics, 2016).

  3. 3. Additionally, the authors overlook recent advancements in rheological modeling of blood, such as K. Giannokostas et al.'s research on advanced constitutive modeling of thixotropic elasto-visco-plastic behavior of blood (Materials, 2020) and AN Beris et al.'s review on recent advances in blood rheology (Soft Matter, 2021). No reference is made.

  4. 4. An important concern raised by the authors is regarding the rigid wall assumption leading to overdetermination of wall shear stress (WSS). However, the authors utilize an inelastic model known to overdetermine WSS, lacking consideration for the elastic effect of blood and the streamwise normal stress component. Studies like K. Giannokostas et al.'s work on quantifying non-Newtonian effects of pulsatile hemodynamics in tubes (Journal of Non-Newtonian Fluid Mechanics, 2021) emphasize this aspect. Additionally, the model used fails to capture kinematic phase lag, a crucial parameter in transient calculations.

  5. 5. The absence of reported lip open cavity instability or flow instability over an open cavity raises questions about calculation convergence. Further investigation into this aspect is warranted, as addressed in studies like K. Giannokostas and Y. Dimakopoulos's TEVP model predictions of pulsatile blood flow in 3D aneurysmal geometries (Journal of Non-Newtonian Fluid Mechanics, 2023).

  6. 6. The tissue modeling is very simple since it doesnot account for the multilayer structure and the anistropicity of the tissue.
Comments on the Quality of English Language

Only minor errors that do not affect the quality of the manuscript.

Reviewer 3 Report

Comments and Suggestions for Authors

This study evaluates the effect of vessel wall modeling on intracranial aneurysm hemodynamics. A novel FSI modeling is proposed, and the results are compared to the rigid wall models. The results suggest that the FSI model can lead to different hemodynamic results than the rigid wall model, and the significance of the difference varies for different aneurysmal geometry. 

This study is designed properly. The manuscript is well written, with a sufficient introduction to the background and previous works, clear explanations of the methods, detailed descriptions of the results, and sufficient discussion. There are only a few minor suggestions from the reviewer:

1. On some figures, the text location seems to be off. For example, the text annotations for the colorbars are too far from the colorbars. Please correct.

2. It'll be beneficial to add the time-series analyses of the WSS to figure 5, similar as the analyses of Vy and pressure. The WSS can be represented using some statistical value in a region of interest, e.g., the mean WSS on the aneurysmal dome. 

3. I would suggest adding some violin plots to show and compare the statistical distributions of WSS (and other hemodynamic parameters if applicable) between the two models, as a compliment to figures 8 & 9.

Round 2

Reviewer 2 Report

Comments and Suggestions for Authors

It's evident that the authors have taken steps to address the concerns outlined in the previous report. However, it's essential that these adjustments and clarifications are not only provided in their responses but also integrated into their paper (issues 4 & 5). By incorporating these changes directly into their manuscript, the authors can ensure that their paper accurately reflects the modifications made and effectively addresses any issues raised.